# Response and inversion of skewness parameters to meteorological factors based on RGB model of leaf color digital image

**Pei Zhang**[1], **Zhengmeng Chen**[2], **Fuzheng Wang**[3], **Hongyan Wu**[1], **Ling Hao**[4], **Xu Jiang**[5], **Zhiming Yu**[6], **Lina Zou**[2], **Haidong Jiang**[6]*

**1** Jiangsu Meteorological Bureau, Nanjing, China, **2** Longyan Company of Fujian Provincial Tobacco Corporation, Longyan, China, **3** QinGengRen Modern Agricultural Science and Technology Development (Huai'an) Co., Ltd., Huai'an, China, **4** Lianyungang Meteorological Bureau, Lianyungang, China, **5** Tufts University, Boston, MA, United States of America, **6** Key Laboratory of Crop Physiology and Ecology in Southern China, Ministry of Agriculture, Jiangsu Collaborative Innovation Center for Modern Crop Prodution, National Engineering and Technology Center for Information Agriculture, Nanjing Agricultural University, Nanjing, People's Republic of China

* hdjiang@njau.edu.cn

## Abstract

In the natural environment, complex and changeable meteorological factors can influence changes in the internal physiology and phenotype of crops. It is important to learn how to convert complex meteorological factor stimuli into plant perception phenotypes when analyzing the biological data obtained under the natural field condition. We restored the true gradation distribution of leaf color, which is also known as the skewed distribution of color scale, and obtained 20 multi-dimensional color gradation skewness-distribution (CGSD) parameters based on the leaf color skewness parameter system. Furthermore, we analyzed the correlation between the five corresponding meteorological factors and canopy CGSD parameters of peppers growing in a greenhouse and cabbages growing in an open air environment, built response model and inversion mode of leaf color to meteorological factors. Based on the analysis, we find a new method for correlating complex environmental problems with multi-dimensional parameters. This study provides a new idea for building a correlation model that uses leaf color as a bridge between meteorological factors and plants internal physiological state.

## Introduction

Climate fluctuation is one of the main causes of inter-annual variation in food production [1], which is becoming more volatile due to global warming and increasing extreme weather events [2, 3]. Agrometeorologists build various models to monitor crop growth, early warn and evaluate agrometeorological disasters, and forecast crop yield via analyzing a large number of environmental data such as meteorological factors and crop growth and yield through decades [4, 5]. In order to predict and evaluate the results more accurately, crop growth model is widely used as a new technology [6, 7]. It can dynamically simulate the growth process and yield

**Funding:** This work was supported by National Key Research and Development Program of China (2018YFD1000900) and "333 project" research project for high level talent of Jiangsu Province (BRA2019348) (to PZ).

**Competing interests:** The authors have declared that no competing interests exist.

formation of crops [8, 9], with strong mechanism and high level of quantification [10]. The time scale of meteorological factors used in the current crop growth model is relatively large. Also, the phenotypic data of crops used, such as crop phenological period, leaf area index, aboveground biomass and ear quality, and yield, require a certain period of accumulation to change. It is necessary to further improve the plant growth model on a smaller scale in order to discover the impact of environmental changes on crop growth in a timelier manner, especially the occurrence of meteorological disasters.

However, the changes of meteorological factors on the upper scale are complex and volatile, and it is difficult to obtain information on how plants perceive environmental changes under rapidly fluctuating natural conditions through laboratory analysis [11, 12]. As a result, we can better understand the relationship between meteorological factors and crop growth in experiments with limited number and level of meteorological factors under artificial control conditions, but it is difficult to be applied to agricultural production in natural environment [13, 14]. Under natural conditions, the growth, development and yield formation of crops are the result of the interaction of many climatic and management factors [15–17], which is manifested as multi-dimensional features. This feature makes it difficult to clearly determine the internal mechanism and contribution degree of climate change on the crop phenotype, which increases uncertainty in climate change impacts and projections [18]. We must consider how to use the natural environment for research [19–24], making it important to find phenotypic parameters that are readily available and more sensitive and quickly reflect complex environmental changes.

Digital color images contain abundant plant morphology, structure, and color information, and the color of plant leaves can reflect its growth state to some extent [25, 26]. The RGB model is the most commonly used color representation for digital images, which parameters can approximately describe the color depth of plant leaves [27]. Drastic changes in meteorological factors can cause physiological and biochemical reactions in plants, which are reflected in the changes of leaf color information in the visible light images of plants [28]. Particularly when plant leaves display obvious senescence and color change after a long period of severe adversity and drastic climate change [29], the RGB model is used to determine the levels of plant fitness and damage [30]. Using RGB model to analyze the color of leaves has been used to study the chlorophyll content [31], nutrition status [32], and stress level [33], while the relationship between the physiological indexes of different levels and the corresponding leaf color depth parameters was found by designing different treatment levels. However, the traditional RGB model parameters cannot more accurately describe small changes in leaf color, such as the overall distribution characteristics of color. Therefore, when studying the relationship between leaf color and physiological problem such as chlorophyll content and nutrition status, traditional parameters can only be used as image parameters and combined parameters are constructed as leaf color phenotypes [34–36]. Previously, we found that the distribution of tobacco leaf color gradation follows a skewed distribution. By skewed analysis, the RGB model parameters are expanded from the mean to the mean, median, mode, skewness and kurtosis, and the number of parameters is also expanded from 4 to 20. These color gradation skewness-distribution (CGSD) parameters can describe leaf color information more accurately and comprehensively [37], we used (CGSD) parameters to construct a correlation model between leaf color and chlorophyll content in tobacco [37] and between leaf color and yield in soybean [38]. We found that CGSD parameters can be used directly as phenotypic parameters.

In order to study whether the multi-dimensional characteristics of skewed parameters are also suitable for the studies of externally complex meteorological factors, we found images of peppers in the greenhouse and cabbages in the natural environment and correspondent meteorological data from the microclimate observation database constructed by the Jiangsu

Provincial Meteorological Bureau. We analyzed the relationship between color changes in images of pepper canopy and meteorological factors in the environment. Also, we used skewed parameters and meteorological factors such as temperature, humidity, and air pressure and constructed the leaf color response model and the meteorological factor inversion model. Additionally, we used the cabbage in the whole natural environment to verify the similar relationship and models. Our investigation provides new ideas and methods for building the correlation model between meteorological factors and the internal physiological state of plants via leaf color as a bridge.

## Materials and methods

### Pepper image collection in a greenhouse

The pepper was planted in the greenhouse which located at the institute of vegetable crops, Jiangsu academy of agricultural sciences, Nanjing City, Jiangsu Province, China($25°5'$N, $116°58'$E)in 2021.Four plots were set up for repeated treatment. The monitoring camera used for image acquisition (model: DH-SD-65F630U-HN-Q; manufacturer: Zhejiang Dahua Technology Co., Ltd, China) had an image resolution of $1920\times1080$, and the surveillance camera was 360 cm tall when installed. Additionally, the study used fixed focal length shooting and automatic white balance. The observation time was from June 13, 2021 to June19, 2021, in a total of 7 days. Each camera took a photo once every punctual time from 7:00 to 18:00, and the resulting digital photo size was $2560\times2592$. The digital images of pepper canopy at 12 time moments were collected. Finally, 35 images taken at 7:04 AM, 9:04 AM, 12:04 AM, 3:04 PM and 6:05 PM were selected for analysis.

### Pakchoi image collection in a natural environment

Color images of pakchoi were collected in a natural environment at the agricultural demonstration base of Qihu Village, Chahe Town, Hongze County, Huai'an City, Jiangsu Province, China ($119°5$'E, $33°17$'N). The monitoring camera was the same as that used in pepper image acquisition. The shooting time was from December 1, 2018 to February 21, 2019 at 8:06 AM, 9:03 AM, 10:03 AM, and 4:03 PM. In this study, to accurately obtain the cumulative distribution information of pakchoi canopy gradation, we discarded those images with strong light, frost and snow cover when making our selections (**S1 Fig in S1 File**). In the end, 116 canopy images were available for the objects of the verification study. Among them, there were 37 images at 8 AM, 28 images at 9 AM, 27 images at 10 AM, and 24 images at 4 PM.

### Meteorological data acquisition

Meteorological data were obtained from the microclimate observatory in the greenhouse and the national weather station of Hongze, including daily meteorological elements, such as the daily minimum temperature ($T_{dmin}$), daily mean relative humidity ($RH_{dm}$),daily mean atmospheric pressure ($AP_{dm}$), daily mean dew point temperature ($TD_{dm}$)and daily mean vapor pressure ($VP_{dm}$), and hourly meteorological elements, such as the hourly temperature($T_h$), hourly relative humidity ($RH_h$), hourly atmospheric pressure ($AP_h$), hourly dew point temperature ($TD_h$), and hourly vapor pressure ($VP_h$). The data were obtained from the Jiangsu Meteorological Information Center of China.

### Processing and information collection of the image

We refer to Chen's method [37] for processing and collecting information of the image.

**Cutting and denoising of the image.** In this study, Photoshop software (San Jose, CA, USA) was primarily used to preprocess the leaf image, and the leaf background was removed and saved as a PNG image mode (**S2 Fig in S1 File**).

**Information collection of the RGB image.** MATLAB2016R software (referred to as MATLAB [Math Works, Natick, MA, USA]) was used to extract and analyze the RGB images.

**Creation of different color channels gradation array.** First, the *imread* function was used to read each color image. Then the *rgb2gray* function was used to obtain its red, green, blue channels and gray-level image color gradation array. After that, the *double* function was used to convert the color gradation array to a double precision array.

**Construction of color gradation cumulative histogram of the image.** The *imhist* function was used to obtain the color gradation cumulative histogram of red, green, and blue channels, as well as gray-level images, of canopy RGB images, while the *plot* function was used to obtain cumulative curve of each images.

**Establishment of leaf color gradation skewness-distribution (CGSD) parameters table.** The *mean*, *median*, *mode*, *skewness* and *kurtosis* functions were used to analyze the leaf color gradation skewed-distribution characteristics [37], respectively. We obtained 20 CGSD parameters: $R_{Mean}$, $R_{Median}$, $R_{Mode}$, $R_{Skewness}$, $R_{Kurtosis}$, $G_{Mean}$, $G_{Median}$, $G_{Mode}$, $G_{Skewness}$, $G_{Kurtosis}$, $B_{Mean}$, $B_{Median}$, $B_{Mode}$, $B_{Skewness}$, $B_{Kurtosis}$, $Y_{Mean}$, $Y_{Median}$, $Y_{Mode}$, $Y_{Skewness}$ and $Y_{Kurtosis}$, including the mean, median, mode, skewness and kurtosis for the red, green, blue channels, as well as the gray-level image, for each canopy image, respectively. These parameters can describe not only the depth of canopy color but also its distribution. Finally, the CGSD parameters tables of color gradation distribution of RGB images were formed.

## Array distribution normality testing

To detect whether RGB color distribution of RGB images follow a skewed distribution, the *lillietest* and *jbtest* functions were used to conduct the Lilliefors and Jarque-Bera tests of normal distribution for the color gradation distribution of red, green and blue channels, as well as gray-level images, of RGB images.

## Correlation analysis of 20 CGSD parameters to hourly meteorological factors

A correlation analysis with SPSS was used to analyze the relationship between 20 CGSD parameters of leaf RGB images collected at approximately 4 time points (8 AM, 9 AM, 10 AM, and 4 PM.) and 5 time points (7 AM, 9 AM, 12 AM, 3 PM and 4 PM.) each day, and the corresponding hourly meteorological factors ($T_h$, $RH_h$, $AP_h$, $TD_h$ and $VP_h$), with double tail inspection were collected for significant examination.

**Prediction models building and prediction accuracy analysis.** By using SPSS, the linear prediction models of 20 CGSD parameters of leaf RGB images were established by a regression approach based on the *least-square* method, with hourly meteorological factors ($T_h$, $RH_h$, $AP_h$, $TD_h$ and $VP_h$) as the independent variables. Correspondingly, the linear prediction models of meteorological factors based on 20 CGSD parameters were also established by a stepwise regression on the *least-square* method.

The results were saved in MS Excel (Redmond, WA, USA) software, and the prediction accuracy was calculated. The formula is as follows:

Predictive accuracy $= (1 - |$ predictive value $-$ measured value$|/$measured value$) \times 100\%$

**Classification of samples based on meteorological factors.** Considering that the response and inversion relationship of leaf color and meteorological factors are different during different temperature ranges, the cabbage samples (n = 116) were divided into two categories by K-Means cluster analysis via SPSS, by considering temperature as classification factor. After having counted the temperature intervals of the two categories of samples, we consider the category with a higher temperature range as normal temperature group, which is represented by T1. While the lower temperature range is called the low temperature group, which is represented by T2. Unsorted samples are denoted by T0.

A bubble chart for air temperature classification of cabbage samples was made by the ggplot2 function package in R language. The bubble chart took the dew point temperature as the X-axis, the relative humidity as the Y-axis, the air temperature as the bubble size, and the color as the classification mark.

## Results

We selected greenhouse-grown peppers and open-air cabbage from image database from Jiangsu Meteorological Bureau and analyzed the RGB color distribution of their canopy images. A histogram analysis of the cumulative distribution of color gradation was performed on canopy images of pepper and cabbage. The red, green, blue channels, as well as the gray-level image all showed a skewed distribution which was further confirmed by Lilliefors and Jarque-Bera normality tests (**S1 Table in S1 File**). Using the analysis of skewed distribution in MATLAB, we obtained 20 color gradient skewed distribution (CGSD) parameters: $R_{Mean}$, $R_{Median}$, $R_{Mode}$, $R_{Skewness}$, $R_{Kurtosis}$, $G_{Mean}$, $G_{Median}$, $G_{Mode}$, $G_{Skewness}$, $G_{Kurtosis}$, $B_{Mean}$, $B_{Median}$, $B_{Mode}$, $B_{Skewness}$, $B_{Kurtosis}$, $Y_{Mean}$, $Y_{Median}$, $Y_{Mode}$, $Y_{Skewness}$, $Y_{Kurtosis}$, including the mean, median, mode, skewness and kurtosis for four channelsfor each canopy image, respectively.

Plants face changes in temperature, humidity and atmospheric pressure during normal growth. **Fig 1** indicates that there were changes in both meteorological Factors and color of pepper and cabbage canopy throughout a day. And the distribution of RGB color levels vibrated in a certain amplitude in a day.

### The Relationship between the variation of canopy CGSD parameters and meteorological factors

**Pepper in greenhouse.** We analyzed the correlations between CGSD parameters of five time points among 7 days and the meteorological factors of corresponding times. The results showed that most CGSD parameters were extremely significantly related to the relative humidity, water vapor pressure and dew point temperature at the corresponding time (**Fig 2**). Temperature is related to leaf color parameters such as $R_{Mode}$, $G_{Kurtosis}$, and $B_{Mode}$. While the air pressure in the greenhouse is generally stable, and only $B_{Mode}$ responds to it.

**Pakchoi in open air environment.** We analyzed the correlations between CGSD parameter of four time points among all observation days, and the meteorological factors of corresponding times. The results showed that almost all the mean, median, mode, and skewnesss parameters of four channels were extremely significantly related to the five meteorological factors ($T_h$, $RH_h$, $AP_h$, $VP_h$ and $TD_h$), except that $R_{Mean}$, $B_{Mean}$, and $G_{Mode}$ were not correlated with timed temperature (**Fig 3**). The kurtosis of channels was not significantly related to the five meteorological factors with the exception of the kurtosis of G channel, which was extremely significantly related to the dew point temperature and relative humidity at corresponding time.

The correlations between the leaf color parameters of two different crops and environmental factors were quite different.

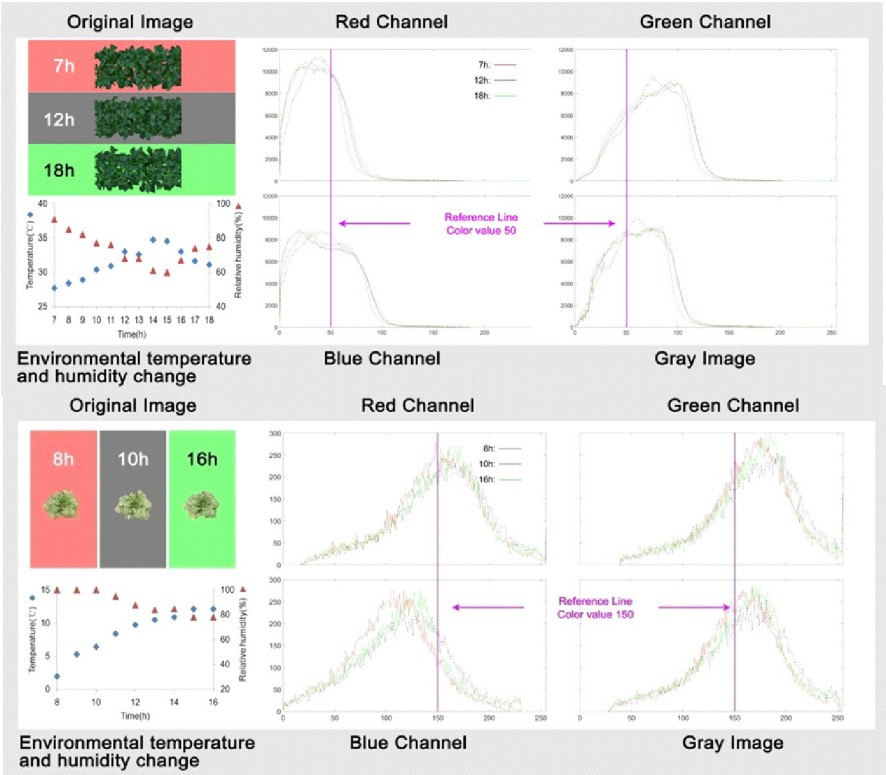

**Fig 1.** The cumulative frequency broken line graph of color gradation distribution of (a) pepper and (b) pakchoi canopy images of three time points in a day. The original image background of pepper canopy at 7 AM, 12 AM, and 6 PM are denoted as red, gray, and green; pakchoi canopy at 8 AM, 10 AM, and 14 PM are denoted as red, gray, and green. The *Plot* function of MATLAB software was used to draw the cumulative frequency broken line graph of color gradation distribution of R, G, B channel, as well as gray image of the three images of pepper canopy. The X-axis is the cumulative frequency, and the Y-axis is the intensity level frequency.

### Canopy color-meteorological response models

**Pepper in greenhouse.**   After clarifying the correlation between canopy CGSD parameters and corresponding hourly meteorological factors, we established canopy color-meteorological response models based on meteorological factors and compared the goodness of fit and prediction accuracy of the models constructed (**Table 1**). Except for B-mean, which cannot be modeled, other 19 skewed parameters all can be used to establish a response model with better fitting effect. According to the F test, the Significance F of 19 response model equation was close to 0, reaching an extremely significant level, indicating that the meteorological factors have significant regressions on pepper canopy color. Based on these results, the prediction of canopy color state can be conducted.

To further verify the accuracy of the model for prediction, we matched the 19 models to the CGSD parameters from prediction samples of 4 repetitions. According to **Table 2**, it can be observed that the canopy color fitting model has a good overall prediction accuracy for the samples of 4 repetitions. The model showed the best prediction accuracy for the mean on each channel, followed by the median, the kurtosis, and the mode, while it was weak for the skewness. With the exception of the mean of B channel, the prediction accuracy for the mean and median on each channel were generally close to or greater than 95%. The accuracy of prediction of the kurtosis on each channel was generally greater than 80%, while the maximum value could be 96.81%. The accuracy of prediction of the mode on each channel was between 64.57%

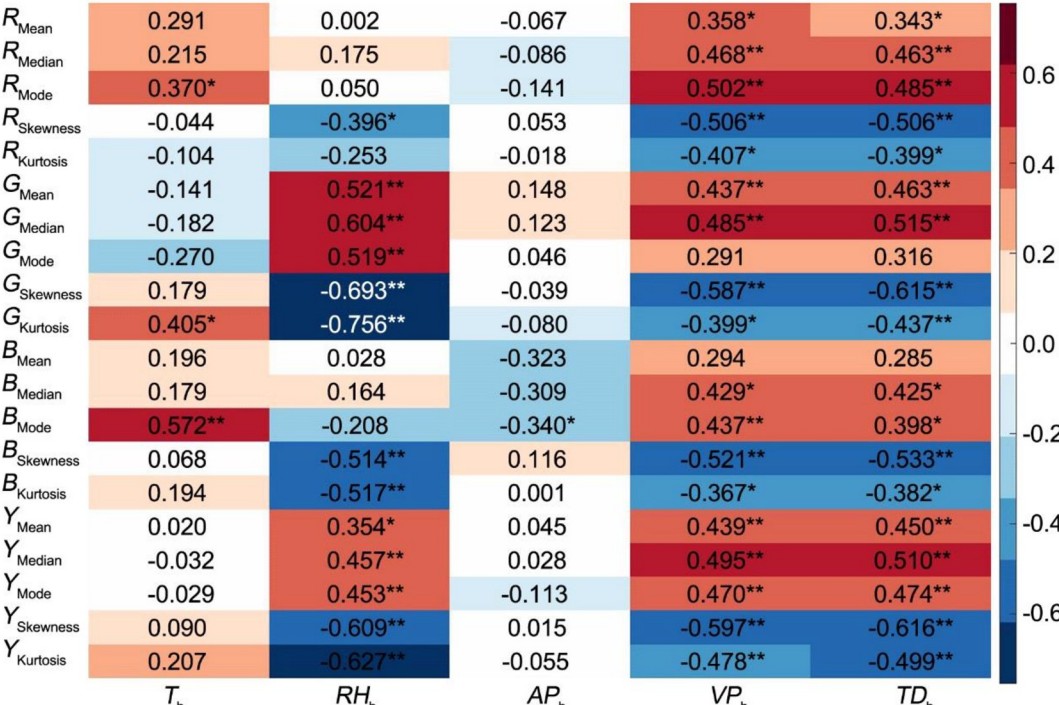

**Fig 2. Correlation coefficient of CGSD parameters of five time points and corresponding hourly meteorological factors in a day.** Correlation analysis of SPSS software was used on 20 canopy CGSD skewness parameters and five hourly meteorological factors ($T_h$, $RH_h$, $AP_h$, $VP_h$ and $TD_h$) of five time points in a day (n = 35). The correlation coefficient obtained by related analysis were drawn to CGSD parameters–daily meteorological factors heat map using the *heatmap* function of MATLAB software. The positive correlation coefficient is shown in red, and the negative correlation coefficient is shown in blue.

- 88.99%, and there were very few outliers. The accuracy of prediction of the skewness was lower between 37.74% - 84.80% with more outliers.

**Pakchoi in open air environment.** Cabbages grow in the open-air environment in winter, whose environmental changes are more severe than that in the greenhouse, and have experienced several cold waves. During the cooling process, the changes of meteorological factors and leaf color are very drastic (**Fig 4** **and S3 Fig in** **S1 File**). As shown in the leaf color response model, there are many outliers of several skewness (**Table 4**). One consideration is that cold wave may be beyond the fluctuation range of the normal growth environment to produce adversity. Additionally, the internal stress response of cabbage caused by the low temperature stress influenced the leaf color. Thus, we classified the meteorological factor.

Taking temperature as the classification factor, the K-M clustering method of SPSS was used to classify the 116 verification samples into two types. The samples of first type was those with the lower limit of temperature of 4.6°C, and the second type was those with the upper limit of temperature of 4.3°C (**Fig 5**).

Canopy color-meteorological response models showed the best prediction accuracy for the mean on each channel, followed by the median, the mode, while it was weak for the skewness. With the exception of the skewness and the kurtosis, the accuracy of prediction of other CGSD parameters was greater than 90%. However, there were a lot of outliers in the model of skewness (**Table 3** **and S2 Table in** **S1 File** **and** **Table 4**).

$B_{Skewness}$, $R_{Skewness}$, and $Y_{Skewness}$ could not be modeled in unclassified (T0) and low temperature (T2) groups, while $B_{Skewness}$, $G_{Skewness}$, and $Y_{Skewness}$ in T1 group cannot be modeled

| | $T_h$ | $RH_h$ | $AP_h$ | $VP_h$ | $TD_h$ |
|---|---|---|---|---|---|
| $R_{Mean}$ | 0.156 | 0.534** | -0.505** | 0.386** | 0.421** |
| $R_{Median}$ | 0.213* | 0.559** | -0.564** | 0.435** | 0.473** |
| $R_{Mode}$ | 0.271** | 0.453** | -0.542** | 0.415** | 0.443** |
| $R_{Skewness}$ | -0.338** | -0.619** | 0.661** | -0.544** | -0.589** |
| $R_{Kurtosis}$ | 0.043 | 0.062 | 0.015 | 0.044 | 0.076 |
| $G_{Mean}$ | 0.283** | 0.700** | -0.539** | 0.569** | 0.604** |
| $G_{Median}$ | 0.320** | 0.707** | -0.576** | 0.595** | 0.631** |
| $G_{Mode}$ | 0.073 | 0.347** | -0.350** | 0.274** | 0.248** |
| $G_{Skewness}$ | -0.344** | -0.663** | 0.593** | -0.575** | -0.621** |
| $G_{Kurtosis}$ | 0.074 | 0.272** | -0.130 | 0.181 | 0.224* |
| $B_{Mean}$ | 0.133 | 0.489** | -0.259** | 0.362** | 0.387** |
| $B_{Median}$ | 0.238* | 0.537** | -0.355** | 0.455** | 0.482** |
| $B_{Mode}$ | 0.410** | 0.498** | -0.426** | 0.541** | 0.569** |
| $B_{Skewness}$ | -0.403** | -0.609** | 0.572** | -0.601** | -0.626** |
| $B_{Kurtosis}$ | 0.001 | -0.120 | 0.082 | -0.094 | -0.068 |
| $Y_{Mean}$ | 0.239** | 0.654** | -0.524** | 0.513** | 0.549** |
| $Y_{Median}$ | 0.288** | 0.672** | -0.573** | 0.553** | 0.590** |
| $Y_{Mode}$ | 0.351** | 0.606** | -0.577** | 0.560** | 0.584** |
| $Y_{Skewness}$ | -0.355** | -0.655** | 0.625** | -0.579** | -0.623** |
| $Y_{Kurtosis}$ | 0.049 | 0.157 | -0.050 | 0.100 | 0.138 |

**Fig 3. Correlation coefficient of CGSD parameters of four time points and corresponding hourly meteorological factors in a day.** Correlation analysis of SPSS software was used on 20 canopy CGSD skewness parameters and four hourly meteorological factors ($T_h$, $RH_h$, $AP_h$, $VP_h$ and $TD_h$) of four time points in a day (n = 116). The correlation coefficient obtained by related analysis were drawn to CGSD parameters–daily meteorological factors heat map using the *heatmap* function of MATLAB software. The positive correlation coefficient is shown in red, and the negative correlation coefficient is shown in blue.

(**Table 3** **and S3, S4 Tables in** **S1 File**). Moreover, in the T1 group, the main factor of the response model was relatively similar among each leaf color parameter, which was similar to the pepper. The main meteorological factors that affect different leaf colors should be discussed.

The forecast accuracy of mean, median and mode are above 90%. However, some kurtosis cannot be modeled, and skewness has many outliers (**Table 4**). In the unclassified (T0) and low temperature (T2) groups, most of the models used two or even three meteorological factors. The outliers of skewness parameter were greatly reduced in the T1 group, which were mainly caused by the low temperature process. The outliers of combined T2 and T1 outliers are also reduced when comparing to T0.

## Meteorological fitting models based on leaf CGSD parameters

**Pepper in greenhouse.** To explore whether changes in meteorological factors can be reflected by leaf color, we also constructed meteorological fitting models based on leaf CGSD parameters of pepper (**Table 5**). By judging the goodness of fit, we found that the multivariate determination coefficient $R^2$ of fitting models equation of $T_h$, $RH_h$, $VP_h$ and $TD_h$ was between 0.433–0.793, indicating that the changes in temperature, relative humidity, vapor pressure and dew point temperature can be reflected by changes in pepper leaf color. And the significance F of the model equation was all close to 0, reaching an extremely significant level. Although the goodness of equation fitting of hourly air pressure was worse than that of the four

**Table 1. Canopy color-meteorological response models of pepper and their goodness of fit (n = 35).**

|  | Models | R-square | Adjusted R-square | RMSE | F value | Significance F |
|---|---|---|---|---|---|---|
| $R_{Mean}$ | Y = 29.094+0.298 $VP_h$ | 0.128 | 0.102 | 2.702 | 4.849 | 0.035 |
| $R_{Median}$ | Y = 22.373+0.462 $VP_h$ | 0.219 | 0.196 | 3.028 | 9.276 | 0.005 |
| $R_{Mode}$ | Y = -22.105+1.687 $VP_h$ | 0.252 | 0.229 | 10.096 | 11.107 | 0.002 |
| $R_{Skewness}$ | Y = 2.465−0.060 $TD_h$ | 0.256 | 0.233 | 0.210 | 11.357 | 0.002 |
| $R_{Kurtosis}$ | Y = 9.624−0.122 $VP_h$ | 0.166 | 0.140 | 0.947 | 6.555 | 0.015 |
| $G_{Mean}$ | Y = 63.842+0.183 $RH_h$ | 0.272 | 0.250 | 3.007 | 12.310 | 0.001 |
| $G_{Median}$ | Y = 46.048+0.214 $RH_h$ +0.705 $TD_h$ | 0.448 | 0.414 | 3.438 | 13.004 | 0.000 |
| $G_{Mode}$ | Y = 33.5+0.745 $RH_h$ | 0.270 | 0.248 | 12.290 | 12.194 | 0.001 |
| $G_{Skewness}$ | Y = 2.069−0.015 $RH_h$ -0.03 $T_h$ | 0.610 | 0.585 | 0.096 | 24.980 | 0.000 |
| $G_{Kurtosis}$ | Y = 3.948−0.014 $RH_h$ | 0.571 | 0.558 | 0.118 | 43.895 | 0.000 |
| $B_{Mean}$ | Unable to modeling |  |  |  |  |  |
| $B_{Median}$ | Y = 31.828+0.379 $VP_h$ | 0.184 | 0.159 | 2.772 | 7.450 | 0.010 |
| $B_{Mode}$ | Y = -46.707+2.392 $T_h$ | 0.327 | 0.306 | 8.273 | 16.012 | 0.000 |
| $B_{Skewness}$ | Y = 1.523−0.026 $TD_h$ -0.005 $RH_h$ | 0.387 | 0.349 | 0.111 | 10.099 | 0.000 |
| $B_{Kurtosis}$ | Y = 4.862−0.021 $RH_h$ | 0.267 | 0.245 | 0.343 | 12.009 | 0.001 |
| $Y_{Mean}$ | Y = 45.433+0.678 $VP_h$ | 0.202 | 0.178 | 2.773 | 8.369 | 0.007 |
| $Y_{Median}$ | Y = 38.077+0.981 $VP_h$ | 0.260 | 0.238 | 3.404 | 11.612 | 0.002 |
| $Y_{Mode}$ | Y = -27.897+3.930 $TD_h$ | 0.224 | 0.201 | 15.042 | 9.552 | 0.004 |
| $Y_{Skewness}$ | Y = 1.722−0.036 $TD_h$ -0.007 $RH_h$ | 0.529 | 0.500 | 0.117 | 17.969 | 0.000 |
| $Y_{Kurtosis}$ | Y = 5.398−0.017 $RH_h$ -0.027 $VP_h$ | 0.465 | 0.431 | 0.239 | 13.898 | 0.000 |

**Table 2. Prediction accuracy analysis of the leaf color-meteorological response models of pepper.**

|  | Repetition 1(Modeling group) | | Repetition 2(n = 35) | | Repetition 3(n = 35) | | Repetition 4(n = 35) | |
|---|---|---|---|---|---|---|---|---|
|  | Number of outliers | Prediction accuracy | Number of outliers | Prediction accuracy | Number of outliers | Prediction accuracy | Number of outliers | Prediction accuracy |
| $R_{Mean}$ | 0 | 94.07% | 0 | 94.57% | 0 | 92.97% | 0 | 94.22% |
| $R_{Median}$ | 0 | 92.80% | 0 | 93.87% | 0 | 90.99% | 0 | 92.22% |
| $R_{Mode}$ | 3 | 71.03% | 1 | 74.10% | 0 | 74.78% | 1 | 69.81% |
| $R_{Skewness}$ | 0 | 82.33% | 1 | 83.73% | 0 | 72.74% | 0 | 80.01% |
| $R_{Kurtosis}$ | 0 | 86.73% | 0 | 80.89% | 0 | 73.84% | 0 | 84.28% |
| $G_{Mean}$ | 0 | 96.92% | 0 | 97.29% | 0 | 94.73% | 0 | 95.13% |
| $G_{Median}$ | 0 | 96.62% | 0 | 96.56% | 0 | 94.38% | 0 | 94.33% |
| $G_{Mode}$ | 0 | 88.99% | 0 | 85.70% | 0 | 88.12% | 1 | 82.71% |
| $G_{Skewness}$ | 11 | 44.44% | 14 | 50.36% | 12 | 37.74% | 10 | 43.85% |
| $G_{Kurtosis}$ | 0 | 96.81% | 0 | 94.17% | 0 | 89.22% | 0 | 93.11% |
| $B_{Mean}$ | Unable to modeling | | | | | | | |
| $B_{Median}$ | 0 | 94.75% | 0 | 94.71% | 0 | 92.06% | 0 | 93.34% |
| $B_{Mode}$ | 1 | 68.89% | 0 | 75.24% | 0 | 64.57% | 0 | 71.28% |
| $B_{Skewness}$ | 1 | 83.90% | 1 | 84.80% | 1 | 74.72% | 2 | 78.89% |
| $B_{Kurtosis}$ | 0 | 90.97% | 0 | 89.29% | 0 | 82.80% | 0 | 88.95% |
| $Y_{Mean}$ | 0 | 96.37% | 0 | 96.86% | 0 | 95.34% | 0 | 95.93% |
| $Y_{Median}$ | 0 | 95.37% | 0 | 95.76% | 0 | 94.33% | 0 | 94.83% |
| $Y_{Mode}$ | 1 | 79.49% | 1 | 72.89% | 0 | 86.27% | 2 | 75.82% |
| $Y_{Skewness}$ | 2 | 72.88% | 1 | 75.31% | 1 | 65.74% | 2 | 71.48% |
| $Y_{Kurtosis}$ | 0 | 93.93% | 0 | 91.06% | 0 | 82.67% | 0 | 91.45% |

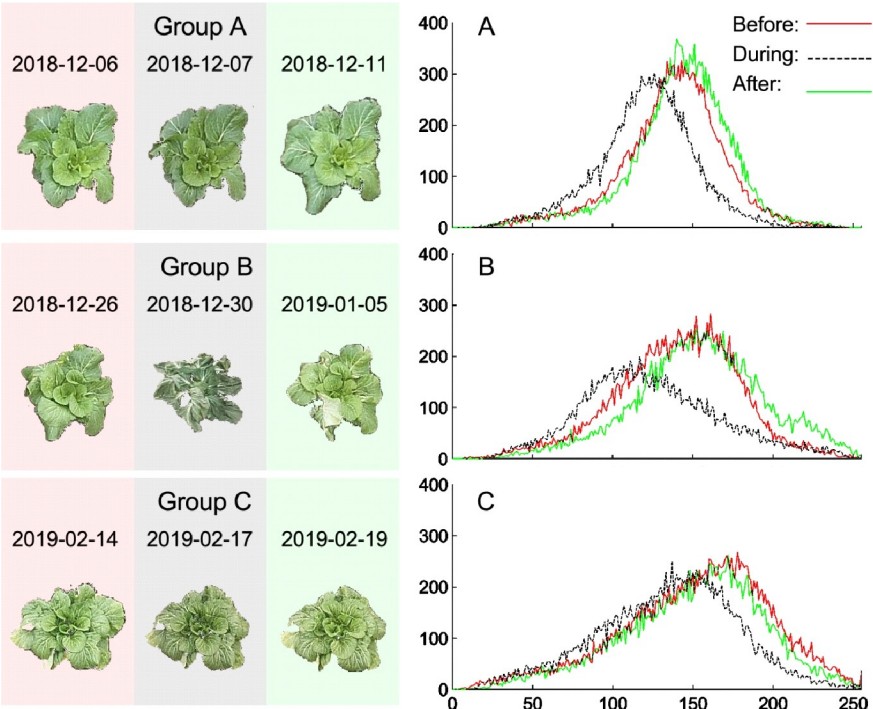

**Fig 4. The cumulative frequency broken line graph of color gradation distribution of R channel of pakchoi canopy images of four cooling processes.** Four cooling processes were selected: Group A-C. In the figure, the original image background of pakchoi canopy before, during, and after cooling processes are denoted as red, gray, and green. The *Plot* function of MATLAB software was used to draw the cumulative frequency broken line graph of color gradation distribution of R channel of left images: before cooling process (Before), during cooling process(During) and after cooling process(After). The X-axis is the cumulative frequency, and the Y-axis is the intensity level frequency.

meteorological factors, the significance F of the model equation was also close to 0.05. All of that indicated that the leaf color parameters of pepper have significant regressions on hourly meteorological factors.

To verify the accuracy of the meteorological fitting model for prediction, we matched the five models to hourly temperature, relative humidity, air pressure, vapor pressure and dew point temperature, from prediction samples of 4 repetitions. According to **Table 6**, it can be observed that the five fitting models had good prediction accuracy which was generally greater than 90% with no outlier. The prediction accuracy of the air pressure fitting model, in particular, was greater than 99%.

**Pakchoi in open-air environment.** Changes in meteorological factors can also be reflected by leaf color as in pepper. Meteorological fitting models were constructed based on leaf CGSD parameters of pakchoi (**Table 7**). By judging the goodness of fit, we found that the significance F of model equation of $T_h$, $RH_h$, $VP_h$ and $TD_h$ was close to 0, reaching an extremely significant level (**S5-S7 Tables in S1 File**). We found that in the unclassified (T0) group, the meteorological fitting models of $T_h$ and $TD_h$ have many outliers (**Table 8**). When we analyzed these outliers, we found that they basically appeared in the cold mornings. Leaf color may experience large fluctuations before and after cold air (**ure 4**) The question is–are these environmental factors beyond the normal growth fluctuation range of cabbage; furthermore, showing abnormal responses in the plant leaf color? After verifying the accuracy of the meteorological fitting model for prediction, 27 outliers were found in hourly temperature and

**Table 3. Canopy color-meteorological response models based on the samples of three types of pakchoi.**

| | Models of T0 (n = 116) | Models of T1 (n = 38) | Models of T2 (n = 78) |
|---|---|---|---|
| $R_{Mean}$ | Y = 942.015+0.466 $RH_h$ -0.813 $AP_h$ -1.024 $TD_h$ | Y = 1041.618–0.876 $AP_h$ | Y = 821.538+3.936 $VP_h$-0.680 $AP_h$ |
| $R_{Median}$ | Y = 1141.802–1.005 $AP_h$ +0.469 $RH_h$ -0.938 $TD_h$ | Y = 1214.356–1.042 $AP_h$ | Y = 1065.848–0.917 $AP_h$ +4.127 $VP_h$ |
| $R_{Mode}$ | Y = 1428.922–1.261 $AP_h$ +0.239 $RH_h$ | Y = 1513.492–1.326 $AP_h$ | Y = 2170.385–1.961 $AP_h$ |
| $R_{Skewness}$ | Y = -18.074+0.018 $AP_h$ -0.005 $RH_h$ | Y = 0.160–0.005 $RH_h$ | Y = -21.535+0.021 $AP_h$ -0.006 $RH_h$ |
| $R_{Kurtosis}$ | Unable to modeling | Unable to modeling | Unable to modeling |
| $G_{Mean}$ | Y = 490.800+0.453 $RH_h$ -0.353 $AP_h$ | Y = 132.454+0.408 $RH_h$ | Y = 124.649+6.961 $VP_h$ |
| $G_{Median}$ | Y = 654.489+0.494 $RH_h$ -0.512 $AP_h$ | Y = 142.005+3.494 $VP_h$ | Y = 671.585+6.362 $VP_h$ -0.525 $AP_h$ |
| $G_{Mode}$ | Y = 2485.302–2.228 $AP_h$ | Y = 91.960+1.217 $RH_h$ | Y = 2708.254–2.441 $AP_h$ |
| $G_{Skewness}$ | Y = -12.610–0.008 $RH_h$ +0.012 $AP_h$ | Y = -0.081–0.050 $VP_h$ | Y = -14.738–0.009 $RH_h$ +0.015 $AP_h$ |
| $G_{Kurtosis}$ | Y = 2.723+0.008 $RH_h$ | Unable to modeling | Y = 2.730+0.112 $VP_h$ |
| $B_{Mean}$ | Y = 90.522+0.276 $RH_h$ | Y = 97.955+1.699 $VP_h$ | Y = 85.599+0.347 $RH_h$ |
| $B_{Median}$ | Y = 86.626+0.346 $RH_h$ | Y = 95.966+2.283 $VP_h$ | Y = 86.809+4.853 $VP_h$ |
| $B_{Mode}$ | Y = 124.269+2.031 $TD_h$ | Y = 96.517+3.907 $VP_h$ | Y = 126.396+2.625$TD_h$ |
| $B_{Skewness}$ | Y = -8.071–0.009 $TD_h$ +0.008P-0.004RH | Y = 0.328–0.050 $VP_h$ | Y = -11.050–0.006 $RH_h$ +0.011 $AP_h$ |
| $B_{Kurtosis}$ | Unable to modeling | Y = 3.542–0.064$T_h$ | Unable to modeling |
| $Y_{Mean}$ | Y = 495.740+0.384 $RH_h$ -0.364 $AP_h$ | Y = 125.832+0.333 $RH_h$ | Y = 116.275+6.339 $VP_h$ |
| $Y_{Median}$ | Y = 675.950+0.419 $RH_h$ -0.539 $AP_h$ | Y = 133.459+2.913 $VP_h$ | Y = 710.198+5.569 $VP_h$ -0.570 $AP_h$ |
| $Y_{Mode}$ | Y = 1133.568+0.481 $RH_h$ -0.983 $AP_h$ | Y = 125.366+4.881 $VP_h$ | Y = 1212.039+6.502 $VP_h$—1.057 $AP_h$ |
| $Y_{Skewness}$ | Y = -14.522–0.007 $RH_h$ +0.014 $AP_h$ | Y = -0.011–0.047 $VP_h$ | Y = -16.961–0.008 $RH_h$+0.017 $VP_h$ |
| $Y_{Kurtosis}$ | Unable to modeling | Unable to modeling | Unable to modeling |

Note: Models of T0 of all samples(n = 116);Models of T1 of the samples of the first type(n = 38); Models of T2 of the samples of the second type(n = 78)

41 outliers in hourly dew point temperature. Further analysis revealed that these outliers occurred during the cooling processes, especially at 8 AM and 9 AM.

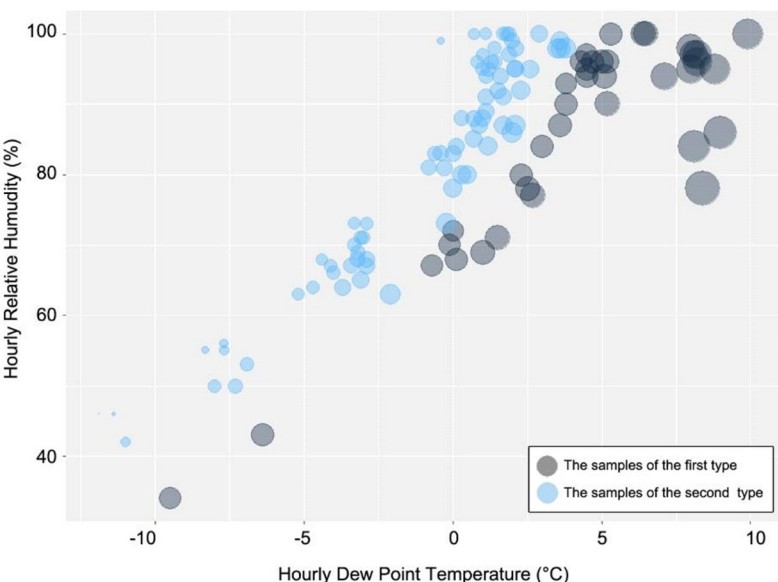

**Fig 5. The cluster bubble map of meteorological factors.** The ggplot2 package of R was used to draw the map. The black bubbles represented the samples of the first type (n = 38, $T_h \geq 4.6$°C), while the blue ones represented the samples of the second type (n = 78, $T_h \leq 4.3$°C). The larger the bubble, the higher the hourly temperature.

**Table 4. Prediction accuracy analysis of the leaf color-meteorological response models of pakchoi.**

| | T0(n = 116) | | T1 (n = 38) | | T2 (n = 78) | |
|---|---|---|---|---|---|---|
| | Number of outliers | Prediction accuracy | Number of outliers | Prediction accuracy | Number of outliers | Prediction accuracy |
| $R_{Mean}$ | 0 | 94.35% | 0 | 94.33% | 0 | 94.12% |
| $R_{Median}$ | 0 | 94.08% | 0 | 94.19% | 0 | 93.58% |
| $R_{Mode}$ | 0 | 91.80% | 0 | 92.47% | 0 | 91.74% |
| $R_{Skewness}$ | 46 | 39.34% | 3 | 67.63% | 35 | 66.53% |
| $R_{Kurtosis}$ | | | | | | |
| $G_{Mean}$ | 0 | 95.75% | 0 | 95.59% | 0 | 95.86% |
| $G_{Median}$ | 0 | 95.43% | 0 | 95.58% | 0 | 95.69% |
| $G_{Mode}$ | 0 | 83.58% | 0 | 86.07% | 0 | 82.45% |
| $G_{Skewness}$ | 59 | 39.33% | 3 | 81.58% | 33 | 35.68% |
| $G_{Kurtosis}$ | 0 | 89.29% | | | 0 | 88.14% |
| $B_{Mean}$ | 0 | 94.59% | 0 | 94.96% | 0 | 95.30% |
| $B_{Median}$ | 0 | 94.08% | 0 | 94.69% | 0 | 94.51% |
| $B_{Mode}$ | 0 | 91.66% | 0 | 91.93% | 0 | 92.46% |
| $B_{Skewness}$ | 59 | 58.87% | 14 | 58.06% | 49 | 47.13% |
| $B_{Kurtosis}$ | | | 0 | 90.81% | | |
| $Y_{Mean}$ | 0 | 95.40% | 0 | 95.46% | 0 | 95.31% |
| $Y_{Median}$ | 0 | 95.12% | 0 | 95.57% | 0 | 95.27% |
| $Y_{Mode}$ | 0 | 93.11% | 0 | 93.56% | 0 | 93.16% |
| $Y_{Skewness}$ | 58 | 47.54% | 3 | 78.26% | 22 | 46.84% |
| $Y_{Kurtosis}$ | | | | | | |

After classifying the samples and rebuilding the models, we found that the outliers of $T_h$ and $TD_h$ are greatly reduced, and their prediction accuracy is improved.

## Discussion

### CGSD parameters can quickly reflect the change of leaf color caused by complex meteorological factors

In this study, we found that the cumulative distribution of leaf color levels of greenhouse-grown pepper and open-air-grown cabbage conformed to a skewed distribution. By analyzing the correlation between the CGSD parameters their canopy leaf color and five meteorological factors, we found that they are closely related. Additionally, most leaf color skewness parameters and meteorological factors all can build leaf color response models and meteorological inversion models. This shows that the canopy color of pepper and cabbage is very sensitive to changes in meteorological factors, which can be quickly described by CGSD parameters. Different plant may have different relationships between leaf color and meteorological factors.

**Table 5. Meteorological fitting models based on leaf CGSD parameters of pepper and their goodness of fit.**

| | Models | R-square | Adjusted R-square | RMSE | F value | Significance F |
|---|---|---|---|---|---|---|
| $T_h$ | $Y = 42.866+0.174\, B_{Mode} -0.232\, G_{Median}$ | 0.495 | 0.463 | 1.739 | 15.677 | 0.000 |
| $RH_h$ | $Y = 166.136-31.418\, G_{Kurtosis} -1.815\, B_{Mean} +1.061\, G_{Median}$ | 0.707 | 0.679 | 5.598 | 24.932 | 0.000 |
| $AP_h$ | $Y = 1003.672-0.108\, B_{Mode}$ | 0.116 | 0.089 | 3.006 | 4.320 | 0.046 |
| $VP_h$ | $Y = 21.555-21.771\, Y_{Skewness} +4.583\, B_{Kurtosis}$ | 0.433 | 0.397 | 2.656 | 12.195 | 0.000 |
| $TD_h$ | $Y = 19.041-13.163\, Y_{Skewness} +2.731\, B_{Kurtosis}$ | 0.455 | 0.421 | 1.543 | 13.376 | 0.000 |

**Table 6. Analytical results of prediction accuracy of meteorological fitting models based on leaf CGSD parameters of pepper.**

| | Repetition 1(Modeling group) | | Repetition 2(n = 35) | | Repetition 3(n = 35) | | Repetition 4(n = 35) | |
|---|---|---|---|---|---|---|---|---|
| | Number of outliers | Prediction accuracy | Number of outliers | Prediction accuracy | Number of outliers | Prediction accuracy | Number of outliers | Prediction accuracy |
| $T_h$ | 0 | 95.42% | 0 | 94.01% | 0 | 87.60% | 0 | 90.14% |
| $RH_h$ | 0 | 94.50% | 0 | 92.16% | 0 | 79.12% | 0 | 91.80% |
| $AP_h$ | 0 | 99.80% | 0 | 99.78% | 0 | 99.75% | 0 | 99.77% |
| $VP_h$ | 0 | 92.74% | 0 | 91.65% | 0 | 92.92% | 0 | 91.30% |
| $TD_h$ | 0 | 94.76% | 0 | 93.78% | 0 | 94.75% | 0 | 93.51% |

The $B_{Mean}$ of pepper leaf color does not change with the five meteorological factors, while only kurtosis among the parameters of cabbage leaf color is not affected by them.

Meteorological factors show periodic changes under natural conditions, manifested as diurnal changes or changes due to weather processes [29, 39]. During these processes, the leaf color waveform would vibrate, which are similar under similar environmental conditions. Peppers in the greenhouse are a good representative in this experiment. However, the cabbage in the open air has experienced a severe cooling process caused by several cold waves. During this process, the movement of waveform is relatively obvious, far exceeding the offset range of the day. When the temperature rises, the waveform gradually recovers, which further indicates the similarity of leaf color waveforms under similar environmental conditions. Also, it shows that the leaf color phenotype responds well to meteorological factors.

## The multidimensionality of CGSD parameters is ideal for describing leaf color phenotypes under the influence of complex meteorological factors

It is very difficult to simulate meteorological factors under experimental conditions due to its complexity and variety [40]. In the natural conditions, crops are faced with constantly changing meteorological factors. The key to obtain biological data under natural field conditions is how to transform the stimulation of real meteorological factors into plant internal physiological parameters or phenotypic parameters [5]. Phenotypic parameters are closely related to internal physiological parameters and are easier to obtain [25]. However, due to the complexity of meteorological factors, the parameter system must be multi-dimensional when building the relationship between meteorological factors and phenotypes [26, 41]. This is why the normal distribution pattern based RGB model cannot satisfy on the. In this study, the normal parameters have only one dimension of mean; however, the meteorological factors cannot be modeled by the mean of each channel. This may be the reason for the data derivation of the

**Table 7. Meteorological fitting models based on leaf CGSD parameters of pakchoi.**

| | Models of T0(n = 116) | Models of T1 (n = 38) | Models of T2 (n = 78) |
|---|---|---|---|
| $T_h$ | $Y = 1.650+0.109 B_{Mode} -0.104 B_{Mean}$ | $Y = -5.810+0.108 B_{Median}$ | $Y = -2.680+0.039 B_{Mode}$ |
| $RH_h$ | $Y = -54.504+1.609 G_{Median}-0.308 R_{Mode}-0.109G_{Mode}-6.654R_{Kurtosis}-0.395B_{Mean}$ | $Y = -21.266+1.471 G_{Median}-1.296 B_{Mean}$ | $Y = -70.290+1.668 G_{Median} -0.420 R_{Mode}-10.586 R_{Kurtosis}-0.154 G_{Mode}$ |
| $AP_h$ | $Y = 1034.582+11.796R_{Skewness}+0.313B_{Mean}-0.256Y_{Median}$ | $Y = 1027.531+13.945R_{Skewness}$ | $Y = 1032.128+15.627R_{Skewness}$ |
| $VP_h$ | $Y = -1.257-3.422 B_{Skewness} +0.046 G_{Median}$ | $Y = -9.883+0.107G_{Median}$ | $Y = -7.490+0.132G_{Median}-0.014 G_{Mode}-0.812 Y_{Kurtosis}-0.023 R_{Mode}$ |
| $TD_h$ | $Y = -34.347+0.206 G_{Median}$ | $Y = -30.291+0.201G_{Median}$ | $Y = -38.502+0.384 G_{Median}-0.048 G_{Mode}-2.063 G_{Kurtosis}-0.059 R_{Mode}$ |

**Table 8. Analytical results of prediction accuracy of meteorological fitting models based on leaf CGSD parameters of pepper.**

| | T0(n = 116) | | T1 (n = 38) | | T2 (n = 78) | |
|---|---|---|---|---|---|---|
| | Number of outliers | Prediction accuracy | Number of outliers | Prediction accuracy | Number of outliers | Prediction accuracy |
| $T_h$ | 27 | 62.97% | 0 | 80.70% | 10 | 68.91% |
| $RH_h$ | 2 | 91.56% | 1 | 91.52% | 0 | 91.34% |
| $AP_h$ | 0 | 99.66% | 0 | 99.70% | 0 | 99.67% |
| $VP_h$ | 0 | 73.83% | 2 | 88.43% | 0 | 87.85% |
| $TD_h$ | 41 | 46.82% | 5 | 69.90% | 27 | 57.62% |

mean values from each channel in the traditional RGB model research, while it may bring about the collinearity issues.

We corrected the way of estimating leaf color via normal distribution and its parameter system used in traditional research in our previous studies. Also, we restored the true gradation distribution state of leaf color, which is the skewed distribution pattern of gradation. Then we obtained 20 CGSD parameters based on leaf color skewness parameter system [37]. CGSD parameters is a set of multi-dimensional parameters, which can describe leaf color from five dimensions systematically and comprehensively [35]. Among five dimensions, the mean is described from the mean value of the color scale of each pixel point. The median is described from the median value of the percentile of the distribution of the number of pixels. The mode sums up the color scale with the most frequency of the same color in the digital pixel. Skewness and kurtosis perform high-level calculations on the skewness and focus of color scale distribution to achieve quantitative description based on the concept of matrix. The multi-dimensionality of CGSD parameters enables them to have a basis for building relationships with complex meteorological factors.

## The potential applications of CGSD parameters in predicting early warning of disaster

The magnitude of changes in meteorological factors also affected leaf color phenotypes [28]. The meteorological factors, such as air pressure, in the glass greenhouse are not very drastic. As a result, the peppers grown there have not experienced drastic weather changes. This leads to the fact that both leaf color skew parameters and meteorological factors can build leaf color response models and meteorological inversion models with high prediction accuracy and without outliers. In contrast, the meteorological factors change much more drastically in the open-air environment, especially in winter or summer. Thus, there are a large number of outliers appeared when building a model for open-air grown cabbage with all process time points. This may due to the fact that the meteorological factors at some time points exceeded its normal growth range, resulting in adversity stress. This may further trigger the internal stress response of the plant; therefore, aggravating the change of leaf color. The results of clustered analysis of meteorological factors proved our assumption. We cannot use the same model to describe the relationship between meteorological factors and canopy leaf colors of cabbage grown in the open air during the entire process compared to peppers grown in a greenhouse. It is necessary to reduce the proportion of outliers through classification modeling; therefore, improving the prediction accuracy. The lower and limit of temperature of 2 types of samples are near to the biological zero of cool season crops. This phenomenon may become a new way to study the threshold of crop adversity.

Abiotic stresses such as high temperature, low temperature, drought and flood are mainly caused by changes in meteorological factors, and even biological disasters such as insect pests

and diseases are also affected by meteorological factors. With the help of the correlation model between plant leaf color phenotypes and meteorological factors, it is possible to realize the monitoring, early warning and prevention of meteorological disasters under drastic environmental changes by monitoring changes in meteorological factors and leaf color phenotypes.

## Conclusion

In conclusion, multi-dimensional plant phenotypes, such as CGSD skew parameters of leaf color, can well respond and invert to meteorological factors. This makes it possible to use leaf color information as a bridge to construct the correlation model between the external environment and the internal physiological state of the plant. The research studies the relationship between the rapidly changing but mild environmental factors and the internal physiological state of plants, which will provide a new idea for the construction of crop growth models. Also, the research studies the inversion model of CGSD parameters of leaf color to some meteorological factors in the process of large-scale changes, making it possible to analyze the threshold value of abiotic adversity.

## Supporting information

**S1 File. Contains supplementary figures and tables.**
(DOC)

## Author Contributions

**Conceptualization:** Pei Zhang, Zhengmeng Chen, Fuzheng Wang, Haidong Jiang.

**Data curation:** Zhengmeng Chen, Hongyan Wu, Ling Hao, Xu Jiang, Zhiming Yu, Lina Zou.

**Formal analysis:** Zhiming Yu, Lina Zou.

**Funding acquisition:** Pei Zhang, Haidong Jiang.

**Investigation:** Ling Hao.

**Methodology:** Zhengmeng Chen.

**Software:** Xu Jiang.

**Writing – original draft:** Pei Zhang, Zhengmeng Chen.

**Writing – review & editing:** Haidong Jiang.

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
