## [Decision Letter · Decision Letter 0]

22 May 2023

PONE-D-23-03750Response and inversion of Skewness parameters to meteorological factors based on RGB model of leaf color digital imagePLOS ONE

Dear Dr. Zhang,

Thank you for submitting your manuscript to PLOS ONE. After careful consideration, we feel that it has merit but does not fully meet PLOS ONE’s publication criteria as it currently stands. Therefore, we invite you to submit a revised version of the manuscript that addresses the points raised during the review process.

We look forward to receiving your revised manuscript.

Kind regards,

Sathishkumar V E

Academic Editor

PLOS ONE

Journal Requirements:

"This work was supported by  National Key Research and Development Program of China (2018YFD1000900) and "333 project" research project for high level talent of Jiangsu Province (BRA2019348） (to PZ)."

"This work was supported by  National Key Research and Development Program of China (2018YFD1000900) and "333 project" research project for high level talent of Jiangsu Province (BRA2019348） (to PZ)."

6. Thank you for stating the following in your Competing Interests section:  

"NO authors have competing interests"

7. In your Data Availability statement, you have not specified where the minimal data set underlying the results described in your manuscript can be found. PLOS defines a study's minimal data set as the underlying data used to reach the conclusions drawn in the manuscript and any additional data required to replicate the reported study findings in their entirety. All PLOS journals require that the minimal data set be made fully available. For more information about our data policy, please see http://journals.plos.org/plosone/s/data-availability.

8. We note that Figures 1, 4, S1, S2 and S4 in your submission contain copyrighted images. All PLOS content is published under the Creative Commons Attribution License (CC BY 4.0), which means that the manuscript, images, and Supporting Information files will be freely available online, and any third party is permitted to access, download, copy, distribute, and use these materials in any way, even commercially, with proper attribution. For more information, see our copyright guidelines: http://journals.plos.org/plosone/s/licenses-and-copyright.

a. You may seek permission from the original copyright holder of Figures 1, 4, S1, S2 and S4 to publish the content specifically under the CC BY 4.0 license. 

Reviewers' comments:

Reviewer's Responses to Questions

**Comments to the Author**

1. Is the manuscript technically sound, and do the data support the conclusions?

Reviewer #1: Yes

Reviewer #2: Yes

Reviewer #3: Partly

2. Has the statistical analysis been performed appropriately and rigorously? 

Reviewer #1: Yes

Reviewer #2: Yes

Reviewer #3: Yes

3. Have the authors made all data underlying the findings in their manuscript fully available?

Reviewer #1: Yes

Reviewer #2: Yes

Reviewer #3: No

4. Is the manuscript presented in an intelligible fashion and written in standard English?

Reviewer #1: Yes

Reviewer #2: Yes

Reviewer #3: Yes

5. Review Comments to the Author

Reviewer #1: The results from this manuscript are impressive. In this study, the author found the closely related relationship between the leaf color levels with some meteorological factors, built the response models. They also drawn the conclusions that the multidimensionality of CGSD parameters is ideal for describing leaf color phenotypes under the influence of complex meteorological factors, and the CGSD parameters is the potential applications in predicting early warning of disaster. this paper may be useful for modern agricultural production. I thought this manuscript should be accepted and published.

Reviewer #2: The ms quantified the skewed distribution of color scale, and obtained the relationship between 20 multi-dimensional CGSD parameters based on leaf color skewness parameter system and the five corresponding meteorological factors.The work reads well, the problem and the objectives are well stated in the introduction. The discussion is well organized and to the point, and the conclusions are based on what is found in the results. They can also be useful elsewhere wherever the canopy photos of the crops is taken. Summarizing, the work fits scientific standards and is suitable for journal publication. There are some issues:

1. The meteorological factors daily mean relative humidity (RHdm), daily mean dew point temperature (TDdm)and daily mean vapor pressure (VPdm) are all the discreption of the vapor of the environment, thus, there will be collinearity among them. The equations in Table 1 and 3 should check the collinearity of the dependent factors.

2. in abstract, phenotype change to phenotypes (Line 16), on the leaf color skewness( Line 19), a greenhouse (Line 21), and plants (Line 26)

Reviewer #3: The manuscript as a whole is not up to journal standards, lacks innovation and looks more like a workflow than a scientific study.

The specific problems are as follows:

1. The CGSD abbreviation appears in line 19 for the first time without a full name

2 Line 65 says something about using RGB images to invert chlorophyll content and nutrient status but it doesn't introduce the corresponding literature method. Right

3 The capture of the images of the peppers in the greenhouse was explained how it was done but there was no mention of whether all the images were used in the experiment or the amount of data that was used in the experiment

4 After 120 lines: used some matlab functions such as 127 lines 142 lines, etc. It just say that they use these functions but they don't tell you exactly what they do and they don't tell you what the corresponding functions are. Right

5 There are 20 different parameters in line 138 but it doesn't tell you what each of them means or even what does Y mean

6 In line 161, it is mentioned that there is a preliminary classification of images by K-means, but what is the purpose of classification not introduced in the article

7 There is no comparison with other models throughout the paper. Besides, the paper adopts a linear model and does not point out why the linear model is adopted (line 152).

8 Line 198 "The results showed that almost all CGSD parameters were extremely significantly related to the relative humidity, water vapor pressure and dew point temperature at the corresponding time." From the heat map, it's not extremely significant. Why only one heat map for 5 time periods?

9 Why are the cabbages divided into two groups based on temperature.

10:Line 232 "Except for B-mean, which cannot be modeled, other 19 skewed parameters all can be used to establish a response model with better fitting effect." From the results of Table 1, we can see that most R2s are. 0.3, the fitting effect is not good.

11 The application of CGSD parameter in disaster warning should be discussed in detail.

12 There was no obvious change of meteorological factors in greenhouse cultivated pepper, so the applicability of inversion model could be improved by increasing temperature change.

6. PLOS authors have the option to publish the peer review history of their article (what does this mean?). If published, this will include your full peer review and any attached files.

Reviewer #1: No

Reviewer #2: No

Reviewer #3: No

---

## [Author Response · Author response to Decision Letter 0]

16 Jun 2023

Dear reviewers,

Thank you for your comments concerning our manuscript entitled " Response and inversion of Skewness parameters to meteorological factors based on RGB model of leaf color digital image ". Those comments are all valuable and very helpful for revising and improving our paper, as well as the important guiding significance to our researches. The comments have been carefully taken into account and a new revised submission has been uploaded. We highlighted all the altered passages in light yellow. The responses are as follows,

To Reviewer #2: 

1. The meteorological factors daily mean relative humidity (RHdm), daily mean dew point temperature (TDdm)and daily mean vapor pressure (VPdm) are all the discreption of the vapor of the environment, thus, there will be collinearity among them. The equations in Table 1 and 3 should check the collinearity of the dependent factors.

Response: Thank you for your question. When constructing the response model of leaf color to meteorological factors, we carried out collinearity diagnosis, and eliminated the factors with collinearity, and then carried on the next modeling work.

2. in abstract, phenotype change to phenotypes (Line 16), on the leaf color skewness ( Line 19), a greenhouse (Line 21), and plants (Line 26)

Response: Thank you for your suggestion, we have corrected them.

Reviewer #3: 

1. The CGSD abbreviation appears in line 19 for the first time without a full name. 

Response: Thank you for your suggestion, we have added the full name “color gradation skewness-distribution”.

2 Line 65 says something about using RGB images to invert chlorophyll content and nutrient status but it doesn't introduce the corresponding literature method. 

Response: Thank you for your question, we have added the corresponding literature method in lines 66-68。

3 The capture of the images of the peppers in the greenhouse was explained how it was done but there was no mention of whether all the images were used in the experiment or the amount of data that was used in the experiment.

Response: Thank you for your question, we have illustrated the images used for the final analysis in lines 103-104.

4 After 120 lines: used some matlab functions such as 127 lines 142 lines, etc. It just say that they use these functions but they don't tell you exactly what they do and they don't tell you what the corresponding functions are. 

Response: Matlab functions such as imread, rgb2gray, double, imhist and plot functions, etc., are some basic image processing functions. We have reworked the functionality of each function to more clearly express the purpose of the function.

5 There are 20 different parameters in line 138 but it doesn't tell you what each of them means or even what does Y mean.

Response: Thank you for your notice, we have added what each of 20 CGSD parameters means in lines 140-145.

6 In line 161, it is mentioned that there is a preliminary classification of images by K-means, but what is the purpose of classification not introduced in the article. 

Response: The reason has been added in lines 169-172. The K-means was used to classify the 116 verification samples into two types, which better revealing the response and inversion relationship of leaf color and meteorological factors during different temperature ranges.

7 There is no comparison with other models throughout the paper. Besides, the paper adopts a linear model and does not point out why the linear model is adopted (line 152).

Response: Multiple linear regression method is widely used because of its simple principle and convenient use among many modeling methods. The modeling independent variable factors in this paper are relatively simple, which are single leaf color parameters (in response models) or limited meteorological factors (in inversion models). The multiple linear regression method can quickly and accurately find the main factors closely related to the dependent variables, and reveal the interaction effect between meteorological factors and leaf color phenotypes, which is also the focus of this paper. Of course, the construction of the nonlinear model you mentioned is equally important, and we will further improve it in the subsequent research.

8 Line 198 "The results showed that almost all CGSD parameters were extremely significantly related to the relative humidity, water vapor pressure and dew point temperature at the corresponding time." From the heat map, it's not extremely significant. Why only one heat map for 5 time periods? 

Response: In order to reveal the timely response rule of pepper canopy leaf color phenotypes to meteorological factors, the correlation analysis between the image CGSD parameters of pepper of 5 times in 7 days and the corresponding meteorological factors was carried out. From the heat map (Figure 2), 11 CGSD parameters were extremely significantly and 2 CGSD parameters were significantly related to the relative humidity; 13 CGSD parameters were extremely significantly and 5 CGSD parameters were significantly related to the water vapor pressure; 13 CGSD parameters were extremely significantly and 5 CGSD parameters were significantly related to the dew point temperature. So, “the results showed that most CGSD parameters were extremely significantly related to the relative humidity, water vapor pressure and dew point temperature at the corresponding time.” We have changed “almost all” into “most” to describe the response relationship more accurately.

9 Why are the cabbages divided into two groups based on temperature. 

Response: The reason has been explained in lines 263-270. “Cabbages grow in the open-air environment in winter, whose environmental changes are more severe than that in the greenhouse, and have experienced several cold waves. During the cooling process, the changes of meteorological factors and leaf color are very drastic (Figure 4 and Supplementary Figure S4). As shown in the leaf color response model, there are many outliers of several skewness (Table 4). One consideration is that cold wave may be beyond the fluctuation range of the normal growth environment to produce adversity. Additionally, the internal stress response of cabbage caused by the low temperature stress influenced the leaf color. Thus, we classified the meteorological factor”.

10: Line 232 "Except for B-mean, which cannot be modeled, other 19 skewed parameters all can be used to establish a response model with better fitting effect." From the results of Table 1, we can see that most R2s are. 0.3, the fitting effect is not good. 

Response: There are generally three indexes for judging whether the regression model is effective and whether the fitting effect is good or bad. (1) R2 reflects the fitting goodness of the regression model to the observed values. The larger R2 is, the better the regression model is. (2) F test is the test of the significance of the regression model. When the P-value of the F test is less than 0.05, it indicates that the regression model is significant, that is, the regression model is effective. (3) T-test is a test of the significance of each variable in the regression model. If the P-value of the T-test is less than 0.05, it indicates that the significance of each variable in the regression model is significant, that is, every variable in the regression model is effective.

 We carried out collinearity diagnosis, F test and T-test successively when constructing the response models of 20 CGSD parameters to meteorological factors. Taking Gmedian as an example, the p-value of F-test of the regression model and the p-value of T-test of each variable are all less than 0.05, indicating that the regression model we constructed is valid and every variable in it is valid. Although the low R2 of some models only reflects the poor fitting goodness of the models, and the prediction accuracy of these models is indeed poor, this cannot be used as a standard for the effectiveness of the models, but only indicates that there may be more effective and accurate model construction methods. We will focus on the study in the next stage.

11 The application of CGSD parameter in disaster warning should be discussed in detail. 

Response: Thank you for your suggestion. We have added the discussion of the application of CGSD parameter in disaster warning in lines 425-430.

12 There was no obvious change of meteorological factors in greenhouse cultivated pepper, so the applicability of inversion model could be improved by increasing temperature change. 

Response: Thank you for your suggestion. we will pay more attention to the temperature change in the future study. In fact, the temperature change in the greenhouse is not as drastic as that in the nature, but some meteorological factors such as relative humidity, water vapor pressure, etc., are also closely related to temperature factors, and their changes often include temperature changes.

---

## [Decision Letter · Decision Letter 1]

5 Jul 2023

Response and inversion of Skewness parameters to meteorological factors based on RGB model of leaf color digital image

PONE-D-23-03750R1

Dear Dr. Zhang,

We’re pleased to inform you that your manuscript has been judged scientifically suitable for publication and will be formally accepted for publication once it meets all outstanding technical requirements.

Kind regards,

Sathishkumar V E

Academic Editor

PLOS ONE

Additional Editor Comments (optional):

Reviewers' comments:

Reviewer's Responses to Questions

**Comments to the Author**

1. If the authors have adequately addressed your comments raised in a previous round of review and you feel that this manuscript is now acceptable for publication, you may indicate that here to bypass the “Comments to the Author” section, enter your conflict of interest statement in the “Confidential to Editor” section, and submit your "Accept" recommendation.

Reviewer #1: (No Response)

Reviewer #2: All comments have been addressed

2. Is the manuscript technically sound, and do the data support the conclusions?

Reviewer #1: (No Response)

Reviewer #2: Yes

3. Has the statistical analysis been performed appropriately and rigorously? 

Reviewer #1: (No Response)

Reviewer #2: Yes

4. Have the authors made all data underlying the findings in their manuscript fully available?

Reviewer #1: (No Response)

Reviewer #2: Yes

5. Is the manuscript presented in an intelligible fashion and written in standard English?

Reviewer #1: (No Response)

Reviewer #2: Yes

6. Review Comments to the Author

Reviewer #1: I don't have any question. And I think this manuscription is qualified to be accepted and published.

Reviewer #2: The manuscript analyzed the relationship between the picture characteristics and the meteorological factors, and the relationship helps to improve the management for the crops with big leaves. The author responded to all concerns.

7. PLOS authors have the option to publish the peer review history of their article (what does this mean?). If published, this will include your full peer review and any attached files.

Reviewer #1: No

Reviewer #2: No

---

## [Editor Report · Acceptance letter]

11 Jul 2023

PONE-D-23-03750R1 

Response and inversion of Skewness parameters to meteorological factors based on RGB model of leaf color digital image 

Dear Dr. Jiang:

I'm pleased to inform you that your manuscript has been deemed suitable for publication in PLOS ONE. Congratulations! Your manuscript is now with our production department. 

Kind regards, 

on behalf of

Dr. Sathishkumar V E 

Academic Editor

PLOS ONE